# A Comparison of Neuropsychological Outcomes following Responsive Neurostimulation and Anterior Temporal Lobectomy in Drug-Resistant Epilepsy

**DOI:** 10.3390/brainsci13121628

**Published:** 2023-11-24

**Authors:** Carly M. O’Donnell, Christopher Todd Anderson, Anthony J. Oleksy, Sara J. Swanson

**Affiliations:** Department of Neurology, Medical College of Wisconsin, 8701 Watertown Plank Road, Milwaukee, WI 53226, USA

**Keywords:** drug-resistant epilepsy (DRE), cognitive change, neuropsychological outcomes, responsive neurostimulation (RNS), neuromodulation, anterior temporal lobectomy (ATL), memory decline

## Abstract

Neuropsychological outcomes following temporal lobe resection for drug-resistant epilepsy (DRE) are well established. For instance, left anterior temporal lobectomy (LATL) is associated with a greater risk for cognitive morbidity compared to right (RATL). However, the impact of neuromodulatory devices, specifically responsive neurostimulation (RNS), remains an area of active interest. There are currently no head-to-head comparisons of neuropsychological outcomes after surgical resection and neuromodulation. This study reports on a cohort of 21 DRE patients with the RNS System who received comprehensive pre- and post-implantation neuropsychological testing. We compared both cognitive and seizure outcomes in the RNS group to those of 307 DRE patients who underwent LATL (*n* = 138) or RATL (*n* = 169). RNS patients had higher seizure rates pre-intervention. While fewer in the RNS group achieved Class I Engel outcomes compared to the ATL cohorts, RNS patients also showed seizure frequency declines from pre- to post-intervention that were similar to those who underwent resective surgery. Moreover, the RNS and RATL groups were similar in their neuropsychological outcomes, showing no significant cognitive decline post-intervention. In contrast, the LATL group notably declined in object naming and verbal list learning. Direct comparisons like this study may be used to guide clinicians in shared decision making to tailor management plans for patients’ overall treatment goals.

## 1. Introduction

While anti-seizure medications are considered first-line therapy for epilepsy patients, over a third are classified as drug-resistant as defined by the International League Against Epilepsy [1,2,3]. For these patients, surgical resection historically stood as the only remaining standard option. Though a highly effective method of seizure reduction for some eligible patients, one of the established risks of both temporal lobectomy and resection is cognitive decline, particularly in memory [4,5,6]. These deficits are most apparent when the dominant hemisphere—typically the left—is the site of surgical interest [7]. The year 1997 marked the advent of neuromodulatory therapy when the United States Food and Drug Association (FDA) approved vagal nerve stimulation for the treatment of epilepsy [8,9]. Since then, deep brain stimulation (DBS) and responsive neurostimulation (RNS, NeuroPace, Mountain View, CA, USA) have also been FDA-approved as treatment options for patients with drug-resistant epilepsy (DRE).

After the inception of DBS for the treatment of DRE, reports on its potential neuropsychological impacts were mixed. Though initial DBS publications reported that participants showed no significant changes from their cognitive and psychiatric baselines, others suggested that some patients may suffer from a worsening of their pre-implant psychological status, although these negative correlations were not statistically significant [10,11]. After 5 years of follow-up, some DBS studies showed a significant self-reported increase in depression and worsening of memory, but objective measurements in the same report demonstrated the opposite effect [11,12]. In contrast, the initial trial showed no deterioration in the active RNS cohort compared to sham, and even found some significant improvements, though more recent follow-up studies have shown no significant changes in these domains [13,14,15]. Overall, existing neuropsychological outcome data from RNS appears neutral or positive compared to DBS. Some epilepsy practitioners favor RNS over other therapies because of this.

Neuropsychological outcomes following resective epilepsy surgery and neuromodulatory procedures are currently an area of active interest. New studies investigating the neuropsychological impacts of RNS are currently being reported [16]. However, there have been no head-to-head comparisons of RNS neuropsychological outcomes against other surgical interventions for epilepsy including anterior temporal lobectomy (ATL). ATL may be more likely to result in seizure freedom, but it is also associated with a greater risk for cognitive morbidity, and some people do not qualify for ATL due to nonlocalized seizure foci. However, there are some individuals who are candidates for both RNS and ATL and, for these patients, a choice must be collaboratively made with the patient and their physician on which procedure would be the best option for them. Therefore, in this study, we report on a cohort of 21 DRE patients with RNS who received pre- and post-implantation neuropsychological testing and compare them with a database of over 138 left (LATL) and 169 right (RATL) patients. We hope that our results may serve as a reference for clinicians to more easily counsel patients who are candidates for both surgical resection and neuromodulation. This may assist patients in pursuing treatment options that best align with their goals and lifestyle based on a direct comparison of their likely seizure reduction and effects on mood, memory, and quality of life (QoL).

## 2. Methods

### 2.1. Patient Selection and Eligibility Criteria

We performed a single-institution cohort study utilizing data prospectively collected via electronic medical record (EMR, Epic Systems Corporation, Verona, WI) from 1995 to 2023 at a 735-bed tertiary-care academic hospital in Milwaukee, WI. Study data were collected and managed using Research Electronic Data Capture (REDCap) tools hosted at the Medical College of Wisconsin [17]. REDCap is a secure, web-based application designed to support data capture for research studies, providing (1) an intuitive interface for validated data entry; (2) audit trails for tracking data manipulation and export procedures; (3) automated export procedures for seamless data downloads to common statistical packages; and (4) procedures for importing data from external sources.

The data were prospectively entered and manually banked for all epilepsy surgery candidates evaluated in the Comprehensive Epilepsy Surgery Program. For this study, all patients in the database who were 18 years or older, diagnosed with DRE, treated with either RNS or temporal lobectomy, and who had undergone both pre- and post-operative neuropsychological evaluations were included for analysis. Patients without complete testing (both pre- and post-intervention) were excluded. The study protocol was reviewed and approved by the Medical College of Wisconsin’s Institutional Review Board.

### 2.2. Study Variables

Across all three cohorts, patients underwent a battery of neuropsychological tests prior to their respective interventions and again typically 6 to 12 months after they completed their procedures [18]. All testing was administered using standardized procedures by a trained psychometrist and scored using Heaton norms that adjust for age and education.

Seizure frequency was assessed by patient report during their neuropsychological testing or during appointments with their established epilepsy team as a routine part of their clinical care.

The RNS cohort contained patients with electrodes placed both in the thalamus (*n* = 6) and in the cortex or hippocampus (*n* = 15). Given the small sample size, statistical analysis was not performed between these subgroups (Table 1). Of these patients, 38% (*n* = 8) had undergone surgical resection without sufficient improvement in seizure frequency prior to the placement of their devices.

### 2.3. Statistics

All data analysis was conducted using SPSS (version 26) and Microsoft Excel (version 16.78).

Analysis of variance (ANOVA) and chi-square analyses were conducted to examine between-group (RNS, LATL, RATL) differences in continuous and categorical demographic variables including sex, race, marital status, handedness, and education. Both a chi-square test and an ordinal regression were used to examine group differences in Engel seizure outcome classifications.

A one-way ANOVA followed by post hoc tests (Tukey’s Honestly Significant Difference (HSD) test) was performed on all continuous variables including baseline epilepsy characteristics (seizure frequency, age at onset of seizures, duration) and neuropsychological test scores. The assumption of homogeneity of variance was violated on only one measure: the Boston Naming Test (BNT). There were higher variability in change scores in the LATL group. Therefore, for the BNT, a nonparametric Kruskal–Wallis one-way ANOVA was conducted. Pre- to post-intervention change scores were calculated for the cognitive variables (formula: post-intervention score—pre-intervention score = change score) and between-group change scores were examined using ANOVA. The resulting changes in seizure frequency and cognitive functioning post-intervention are described in Section 3.2.

## 3. Results

### 3.1. Patient Demographics

The authors identified 21 RNS, 169 RATL, and 138 LATL patients who met the inclusion criteria for enrollment. This sample provides 80% power to detect a large effect size along with a 5% significance level using a Tukey’s HSD adjustment to account for the smaller n in the RNS group and the larger ns in the ATL groups.

There were no significant between-group differences in the demographic information. Chi-square analyses revealed no differences between LATL, RATL, and RNS groups for sex, handedness, race, and marital status (Table 2).

### 3.2. Preoperative Epilepsy and Neuropsychological Characteristics

Preoperatively, patients in the RNS cohort had significantly higher rates of monthly seizures (62 seizures/month) compared to both the RATL (25 seizures/month, *p* = 0.010) and LATL cohorts (16 seizures/month, *p* = 0.002). Additionally, the RNS cohort was significantly older on average than the LATL cohort at the age at seizure onset (19 vs. 14 years respectively, *p* = 0.047). Preoperatively, RNS patients had significantly lower scores on the Wechsler Adult Intelligence Scale (WAIS-III/WAIS-IV) Working Memory and Processing Speed indexes compared to both ATL groups [19]. The RNS group had a significantly lower WAIS Verbal Comprehension index score, a measure of verbal intelligence, compared to their RATL counterparts (*p* = 0.007). Finally, the Full Scale IQ (FSIQ) for the RNS group was significantly lower than FSIQ in both ATL groups (RNS = 81.84 vs. RATL = 90.64 vs. LATL = 90.07). These results are further detailed in Table 3.

### 3.3. Post-Intervention Epilepsy Outcomes

Chi-square analysis revealed significant group differences in Engel outcome scores (Table 4 and Figure 1). A Class I Engel outcome was defined as 100% seizure freedom (with the exception of auras), Class II was defined as <100% but ≥90% seizure freedom, Class III was defined as <90% but ≥75% seizure freedom, and Class IV was defined by <75% seizure freedom post-intervention. Both RATL and LATL patients were more likely to experience complete seizure freedom (Engel Class I, RATL 72.8%, LATL 67.4%) compared to the RNS cohort (20.8%). The RNS group had significantly more patients with Class III outcomes compared to the RATL and LATL cohorts. The RNS group also had more Class IV outcomes (33.3% of RNS patients compared to 7.1% of RATL and 5.8% of LATL patients). However, the percent reduction in seizure frequency post-intervention for all three cohorts as assessed by ANOVA was insignificant. LATL patients trended toward a higher level of seizure reduction, but this was not statistically significant (80.3% reduction in seizure frequency compared to 63.9% and 60.1% seizure reduction in the RATL and RNS cohorts respectively, *p* = 0.907 and 0.545).

Based on ordinal logistic regression, there are significant differences (*p* = 0.004) in the observed data and the fitted (assumed) model. The parameter estimates indicate that individuals who undergo an LATL or RATL have a significantly greater likelihood of having a better seizure outcome (*p* = 0.001 for both RATL and LATL) compared to the RNS group.

### 3.4. Post-Intervention Neuropsychological Outcomes

Post-intervention, the RNS and RATL cohorts showed no significant changes in object naming as measured by the BNT or on verbal memory as measured by the Buschke Selective Reminding Test (SRT) [20,21]. However, the LATL cohort showed a significant raw score decline in naming on the BNT (*p* < 0.005) and in verbal memory on the SRT (*p* = 0.008) compared to the RNS cohort (Table 5 and Figure 2). The effect size of the BNT is classified as Cohen’s D (0.89–0.92), which is considered large.

While all three cohorts showed an improvement in QoL as assessed by the Quality of Life in Epilepsy-31 (QOLIE-31) post-intervention, both the RATL and LATL groups showed significantly greater improvements than RNS (RNS: 0.308 vs. RATL: 10.013, *p* = 0.003; LATL: 6.712, *p* = 0.050 respectively) [22]. The RATL cohort also had a significantly greater improvement in their QOLIE-31 scores than their LATL counterparts.

Changes in all other neuropsychological scores post-intervention for all three groups were not statistically significant.

**Table 5 brainsci-13-01628-t005:** Neuropsychological change scores across groups post-intervention.

Variables	*p*-Value (RNS vs. RATL)	RATL (*n* = 148–160)	RNS (*n* = 13–21)	LATL (*n* = 119–132)	*p*-Value (RNS vs. LATL)
WAIS					
FSIQ	0.631	0.744	1.750	0.239	0.475
VCI	0.525	1.131 ^a^	0.053	−3.238 ^a^	0.056
PRI	0.424	−0.101	2.158	1.813	0.904
WMI	0.766	0.373	−0.368	−0.439	0.978
PSI	0.893	−0.865	−0.478	1.765	0.459
Other assessments					
BNT	0.453	0.970 ^a^	0.217 ^b^	−5.508 ^a,b^	0.005 *
SRT	0.810	1.354 ^a^	2.278 ^b^	−8.160 ^a,b^	0.008 *
WMS- Visual Reproduction	0.222	−1.500	−0.353	−0.071	0.787
COWA	0.692	−0.051	−1.381	0.025	0.680
Trails B	0.206	1.429	−3.824	1.246	0.228
QOLIE-31	0.003 *	10.013 ^a^	0.308 ^a,b^	6.712 ^a,b^	0.050 *
MMPI—Scale 2	0.547	3.556	3.327	0.750	0.341

* Indicates statistical significance (*p* ≤ 0.05). Values with the same superscript (a or b) are significantly different from each other. RATL = right anterior temporal lobectomy; RNS = responsive neurostimulation; LATL = left anterior temporal lobectomy. WAIS = Wechsler Adult Intelligence Scale III or IV. FSIQ = Full Scale IQ; VCI = Verbal Comprehension Index Score; PRI = Perceptual Reasoning Index Score; WMI = Working Memory Index; PSI = Processing Speed Index. SRT = Buschke Selective Reminding Test [21]; BNT = Boston Naming Test [20]; COWA = Controlled Oral Word Association Test; Trails B = Trail Making Test Part B [23]; QOLIE-31 = Quality of Life in Epilepsy-31 [22]; MMPI-2 = Minnesota Multiphasic Personality Test-Depression Scale.

## 4. Discussion

To our knowledge, this is the first study that directly compares both the neuropsychological and seizure outcomes of DRE patients who underwent traditional resective surgery via ATL or neuromodulatory therapy via the RNS System. These results provide valuable insight into the potential benefits and drawbacks of the myriad surgical treatment options currently available for epilepsy, shedding new light on the impact of devices—particularly the RNS System—on cognitive function and seizure control.

It is evident that patients in the RNS cohort on average had a significantly higher frequency of seizures pre-intervention compared to the two ATL cohorts. As this study was strictly observational, patients were not randomly assigned to treatment groups (i.e., resection vs. RNS). Consequently, the patients who underwent RNS implantation were often clinically selected for this treatment because they had more severe forms of epilepsy than their ATL counterparts—a form of inherent selection bias. In many cases (38% of this cohort), those selected had already failed previous resective interventions as a result of the highly refractory nature of their diseases. Ultimately, fewer patients in the RNS cohort achieved Class I Engel outcomes compared to those who underwent ATL. In other words, the RNS cohort had a significantly lower rate of complete seizure freedom post-intervention. This is an expected finding among those who receive neuromodulatory device implantation for epilepsy treatment. However, this cohort still achieved a reasonable proportion of Class II outcomes, indicating substantial seizure reduction (<100% seizure freedom but ≥90% reduction in seizure frequency). Nearly half (42%) of the RNS group experienced an Engel Class I or Class II outcome. Conversely, the LATL and RATL cohorts achieved a higher percentage of Class I outcomes, demonstrating an overall greater likelihood of complete seizure freedom, but were still not without some Class IV outcomes. Overall, these findings suggest that RNS is an effective option for eligible DRE patients with high baseline seizure frequencies, even though achieving complete seizure freedom may be less likely compared to surgical resection.

Our results also show that the RNS cohort had no significant changes in cognitive measures of intelligence, memory, object naming, verbal fluency, and executive functioning. This is an important finding as it suggests that RNS is not associated with cognitive decline post-intervention, which is consistent with existing studies. Interestingly, the RNS patients, like those in the RATL group, have significantly better cognitive outcomes in object naming and verbal memory than the LATL cohort. This result is consistent with previous literature, which has shown that LATL is associated with greater cognitive morbidity, particularly in language and memory function.

This study also assessed the impact of treatment modalities on the QoL of patients with DRE. Both ATL cohorts showed significantly greater improvements in their QoL as assessed by the QOLIE-31 post-intervention compared to the RNS group. However, considering the persistence of seizures in the RNS group, as evidenced by their higher Engel class assignments, it is possible that their comparatively lower QoL is related to the persistence of seizures rather than the treatment itself. In fact, considerable research supports the finding that seizure freedom is one of the most significant determinants of QoL [24]. Regardless of the underlying cause, this finding is an important consideration when counseling patients, as QoL is a crucial aspect of overall well-being and can significantly influence treatment decisions.

It is important to consider how handedness may affect the results reported in this study. Handedness can be associated with an atypical language representation or reversed language dominance. However, the majority of left-handed individuals are still left-dominant for language. In this study, there was no significant difference in rates of left-handedness across the three groups. The finding of a trending higher rate (that is not statistically significant) of left-handedness in the LATL group makes intuitive sense as disruption of the left hemisphere due to seizures can lead to pathological left-handedness and a reorganization of language to the right hemisphere. While it is more likely that there is an atypical language representation in the LATL group due to their left-sided seizure foci, the reorganization of language to the right hemisphere in this group only makes our findings stronger. To confirm this, we analyzed the two most significant cognitive variables (the BNT and SRT) with all left-handed patients removed from each of the three groups. The results for these tests remained identical with these patients excluded from analysis (*p* < 0.0001).

This study has several notable limitations that warrant consideration. Firstly, its observational design inherently introduces the risk of selection bias, as patients were not randomly assigned to treatment groups. This lack of randomization may have influenced our results, especially given the varied baseline characteristics of individuals in the smaller RNS cohort. The fact that the RNS group has a more severe seizure disorder with more frequent seizures at baseline introduces an unavoidable confounder. Additionally, the relatively small sample size within the RNS group reduces the generalizability of our findings. We acknowledge the need for further studies with larger and more diverse cohorts and with more balanced sample sizes across treatment groups to enhance statistical robustness—particularly given the unequal number of patients across our three groups. Another limitation is the lack of standardization in the placement of RNS electrodes. The varied locations within the thalamus, cortex, and hippocampus could potentially introduce variability in neuropsychological outcomes. Future studies containing larger sample sizes should further subdivide the RNS cohort by their electrode location and specifically by thalamic vs. nonthalamic placement. It would also be beneficial for future investigations to incorporate extended follow-up periods to evaluate the sustainability of both seizure control and neuropsychological outcomes in comparison to ATL. In summary, these limitations underscore the need for cautious interpretation of this study’s findings and emphasize the importance of addressing these challenges in future research endeavors to enhance the validity and applicability of our results.

Ultimately, this study contributes to our understanding of the trade-offs between surgical resection and neuromodulatory therapies such as RNS in the treatment of DRE. While ATL patients appear to achieve comparatively better seizure control and improved QoL, RNS therapy is a reasonable alternative for those with high baseline seizure frequency as it does not result in significant cognitive decline. These results underscore the importance of considering individual patient characteristics and goals when choosing the most suitable treatment approach. As the field of neuromodulation continues to evolve, ongoing research will be vital to further refine our understanding and optimize patient care in the management of drug-resistant epilepsy.

## 5. Conclusions

In appropriately selected patients, RATL and RNS improve epilepsy control and do not appear to result in cognitive decline, while LATL patients may experience reduced function in metrics including object naming and verbal memory. The percent reduction in seizure frequency was similar across groups with all three cohorts experiencing a 60–80% decrease post-intervention. However, the RNS cohort contained fewer patients who ultimately achieved a Class I Engel outcome. This is likely related in part to the RNS cohort’s higher baseline seizure frequency and overall epilepsy burden compared to the ATL cohorts, though it may also be influenced by another as-of-yet unknown factor. Both the RNS and RATL patients show no evidence of cognitive decline 6 months to 1 year postoperatively as assessed by our selected metrics. Conversely, the LATL cohort showed what has been demonstrated in many previous studies: a decline in object naming and verbal memory as measured by word list learning tests.

We hope that our results may serve as an initial reference for clinicians who wish to counsel DRE patients that are candidates for both surgical resection and neuromodulation. This study aims to encourage clinicians to engage in a holistic discussion of surgical treatment options with those living with DRE. Some patients may choose to pursue a path that is more likely to result in complete seizure freedom (ATL), while others may be interested in an intervention that is more likely to preserve or improve their cognitive function as assessed by object naming and verbal memory (RNS as compared to LATL). Guidance from our report is by no means final, as we do not yet have prospective outcome data from clinical trials directly comparing these groups. Nevertheless, this discussion raises the critical topic of patient-centered goals of care. Ultimately, we believe that it is important for patients to be as involved as possible in making decisions regarding their individual treatment plans.

## Figures and Tables

**Figure 1 brainsci-13-01628-f001:**
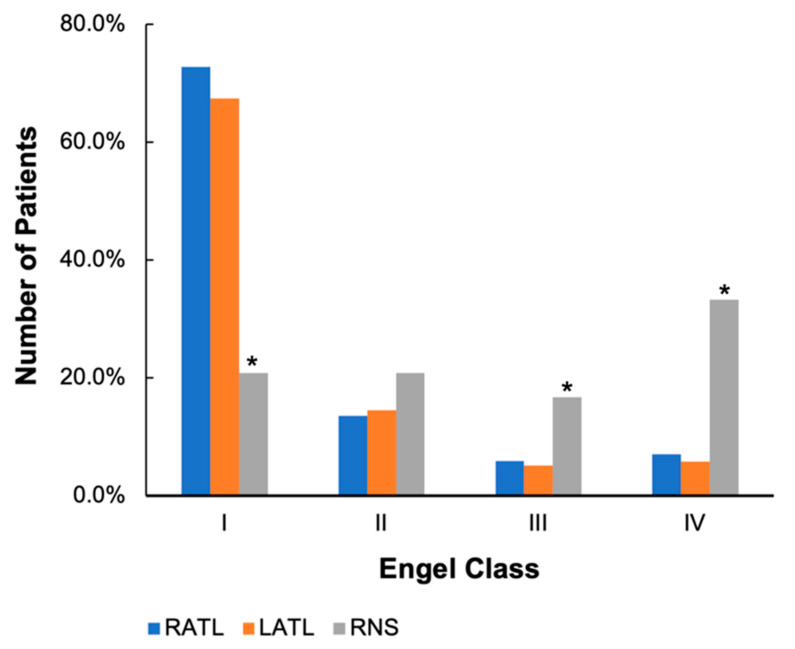
Engel class outcomes. Graph depicts the percent of each cohort represented in the four Engel classes post-intervention. Asterisks represent a statistically significant difference between the RNS cohort and both the LATL and RATL groups.

**Figure 2 brainsci-13-01628-f002:**
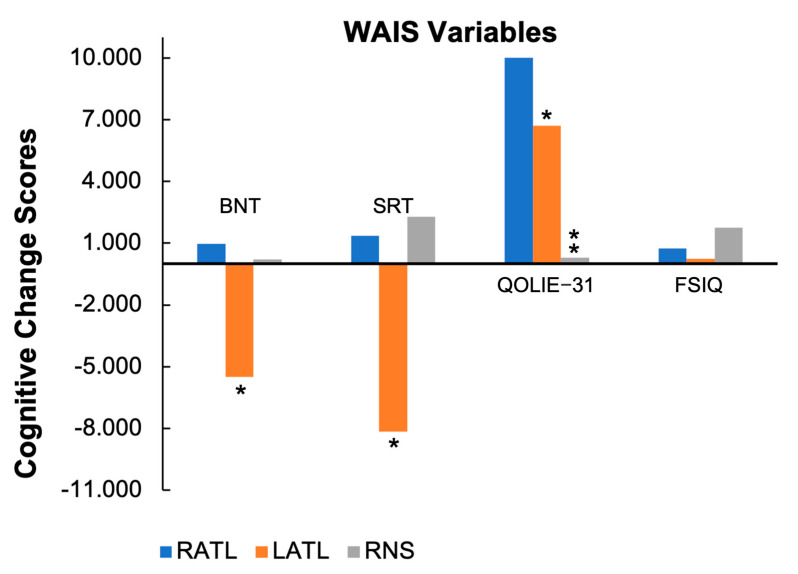
Cognitive change scores. Graph depicts the percent change in each cohort in variables assessed through the WAIS. Asterisks represent a statistically significant difference between the identified group and the other two cohorts.

**Table 1 brainsci-13-01628-t001:** Epilepsy characteristics and surgical history of RNS cohort.

Variables	Thalamic RNS (*n* = 6)	Non-Thalamic RNS (*n* = 15)
Epilepsy diagnosis		
LRE (single focus)	1	14
LRE (multifocal)	1	1
GGE/JME	3	0
LGS	1	0
History of prior resective surgery		
Yes	1	7
No	5	8
Location of RNS electrodes		
LANT	4	0
RANT	4	0
LCM	2	0
RCM	2	0
LFS	1	2
RFS	0	0
LHIP (limbic)	0	11
RHIP (limbic)	0	10
LTS (neocortical)	0	5
RTS (neocortical)	0	0

RNS = responsive neurostimulation. LRE = localization-related epilepsy (also called focal epilepsy); GGE = genetic generalized epilepsy; JME = juvenile myoclonic epilepsy; LGS = Lennox–Gastaut syndrome. LANT = left anterior thalamic nucleus; RANT = right anterior thalamic nucleus; LCM = left centromedian thalamic nucleus; RCM = right centromedian thalamic nucleus; LFS = left frontal cortical strip; RFS = right frontal strip; LHIP = left hippocampal depth strip electrode; RHIP = right hippocampal depth electrode; LTS = left temporal cortical strip; RTS = right temporal cortical strip.

**Table 2 brainsci-13-01628-t002:** Demographic characteristics.

Variables	RATL (*n* = 169)	RNS (*n* = 21)	LATL (*n* = 138)
Sex (%)			
Male	40.2	29.2	48.6
Female	59.8	70.8	51.4
Race/Ethnicity (%)			
White	89.9	83.3	87.7
Black	7.7	12.5	5.8
Hispanic	2.4	4.2	5.1
Asian	0.0	0.0	0.7
Unknown	0.0	0.0	0.7
Marital status (%)			
Married	36.7	37.5	42.0
Divorced	10.7	12.5	9.4
Separated	0.6	0.0	0.7
Never married	48.5	45.8	44.9
Widowed	0.6	4.2	0.0
Unknown	3.0	0.0	2.9
Handedness (%)			
Right	85.8	79.2	81.2
Left	9.5	12.5	16.7
Mixed	4.7	8.3	2.2
Average length of education (years)	12.96	13.29	12.70

All listed variables showed no significant between-group differences. RATL = right anterior temporal lobectomy; RNS = responsive neurostimulation; LATL = left anterior temporal lobectomy.

**Table 3 brainsci-13-01628-t003:** Preoperative epilepsy and neuropsychological characteristics.

Variables	*p*-Value (RNS vs. RATL)	RATL (*n* = 169)	RNS (*n* = 21)	LATL (*n* = 138)	*p*-Value (RNS vs. LATL)
Preoperative epilepsy characteristics					
Seizure frequency (number/month)	0.010 *	25 ^a^	62 ^a,b^	16 ^b^	0.002 *
Seizure frequency excluding simple partials/auras (number/month)	0.003 *	18 ^a^	59 ^a,b^	12 ^b^	0.001 *
Age at seizure onset (years)	0.117	15	19 ^a^	14 ^a^	0.047 *
Age at onset of recurrent seizures (years)	0.441	16	20	17	0.258
Disease duration (years)	0.651	20	21	20	0.668
Age at pre-op neuropsychological testing	0.163	37	41	36	0.080
Preoperative neuropsychological characteristics in intellectual testing (WAIS-III—WAIS-IV)					
Verbal Comprehension Index	0.007 *	92.76 ^a^	83.57 ^a^	89.57	0.083
Perceptual Organizations Index	0.162	93.45	89.00	94.69	0.078
Working Memory Index	0.170	89.41	84.40	91.06	0.071
Processing Speed Index	0.002 *	90.64 ^a^	79.63 ^a,b^	91.68 ^b^	0.002 *
Full scale IQ	0.012 *	90.64 ^a^	81.84 ^a,b^	90.07 ^b^	0.019 *

* Indicates statistical significance (*p* ≤ 0.05). Values with the same superscript (a or b) are significantly different from each other. RATL = right anterior temporal lobectomy; RNS = responsive neurostimulation; LATL = left anterior temporal lobectomy. WAIS = Wechsler Adult Intelligence Scale.

**Table 4 brainsci-13-01628-t004:** Epilepsy outcomes across groups post-intervention.

Variables	RATL (*n* = 153–168)	RNS (*n* = 21)	LATL (*n* = 126–128)
Engel scores (%)			
Class I	72.8 ^a^	20.8 ^a,b^	67.4 ^b^
Class II	13.6	20.8	14.5
Class III	5.9 ^a^	16.7 ^a,b^	5.1 ^b^
Class IV	7.1 ^a^	33.3 ^a,b^	5.8 ^b^
Reduction in seizure frequency (%)	63.9	60.1	80.3

Values with the same superscript (a or b) are significantly different from each other. RATL = right anterior temporal lobectomy; RNS = responsive neurostimulation; LATL = left anterior temporal lobectomy.

## Data Availability

The data presented in this study are available on request from the corresponding author. The data are not publicly available in order to protect the privacy of our patients.

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
