# Peer review of "A Comparison of Neuropsychological Outcomes following Responsive Neurostimulation and Anterior Temporal Lobectomy in Drug-Resistant Epilepsy"

_brainsci, 2023, doi:10.3390/brainsci13121628_

Round 1

Reviewer 1 Report

Comments and Suggestions for Authors

Due to the fact that epilepsy is a chronic (often completely incurable) lifelong disease, the quality of life of patients plays an important role. One of the most important factors determining the quality of life of patients in this group is the presence/absence of cognitive impairment and the ability to control epileptic seizures. It is important to find a balance between these two factors. Therefore, clinicians always have to choose which method of treatment for drug-resistant epilepsy to give preference to. In this regard, the relevance of the manuscript is undoubted. During the audit a number of questions arose:

1) In the “Materials and Methods” section, authors should indicate more precise criteria for inclusion and exclusion from the study.

2) Below Table 2 you should once again give the full explanation of the abbreviations (RATL, RNS, LATL).

3) The authors made interesting findings that with RATL, cognitive functions suffer less than with LATL. The authors explain this fact by interference with the functioning of the left (dominant) hemisphere in right-handed people. However, in Table 2, 16.7% of the subjects were left-handed, which can significantly introduce errors into the results obtained by the authors. The authors need to either completely exclude the group of left-handed patients from this study, or identify additional subgroups (right-handed/left-handed) with additional analysis of the study results obtained in these subgroups. Ambidextra are also of particular interest.

4) The authors did not provide data on the presence/absence of motor deficits in groups (RATL, LATL) after surgical treatment, this in turn may also affect the quality of life of these patients.

Author Response

Thank you very much for your review. Please see our attached response.

Reviewer 2 Report

Comments and Suggestions for Authors

This is an interesting and well written paper. However, I have some comments and suggestions which may improve the quality of this paper.

Introduction

·       The introduction could provide more background on the different surgical treatments for epilepsy (e.g. temporal lobe resection, vagal nerve stimulation, deep brain stimulation). As it stands, the treatments are mentioned briefly without much context.

·       The rationale for comparing RNS to temporal lobe resection specifically could be clearer. Why choose this comparison over other surgical options?

·       The sample sizes between groups are quite uneven, with 21 RNS patients compared to over 300 resection patients. This imbalance should be addressed as a limitation, as it reduces confidence in drawing direct comparisons.

·       Details of the neuropsychological testing methodology are lacking. What specific tests were used? Were they administered in a standardized way across groups? Describe briefly.

Statistics

·       More justification is needed for the use of ANOVA and MANOVA tests. Are the assumptions of these parametric tests met, such as normality and homogeneity of variance across groups? Nonparametric alternatives should be considered if assumptions are violated.

·       The pre-to-post change score analysis could be improved. Using a repeated measures ANOVA may be more appropriate to model the within-subjects pre/post effect over time.

·       Details of the post-hoc tests after the MANOVAs are not provided. Which specific post-hoc tests were used? Were multiple comparisons accounted for?

·       The chi-square test for Engel seizure outcome classifications seems insufficient given that this is an ordered categorical outcome. An ordinal logistic regression may be more appropriate.

·       No effect sizes are reported for the group comparisons. Inclusion of effect sizes would provide more insight into the magnitude of group differences.

·       Statistical power and sample size calculations are not mentioned. Were the samples sizes adequate to detect meaningful group differences on the neuropsychological measures?

·       Multiple comparisons increase risk of Type 1 error. Were any corrections used (e.g. Bonferroni) to control for familywise error rate?

Discussion

·       The rationale for comparing RNS to ATL specifically should be expanded on further. Why was ATL chosen over other resection techniques?

·       The observational study design is a limitation, as patients were not randomly assigned to treatment groups. The potential for selection bias should be addressed.

·       The RNS sample size is small at only 21 patients. This reduces confidence in generalizing the RNS results. A power analysis to determine required sample size should be reported.

·       Details of RNS electrode placement are lacking. Were the placements standardized or individualized? Electrode location could impact outcomes.

·       The results focus on group averages but individual variability in response could be better explored through reporting percentiles or graphs.

·       Conclusions state RNS does not impair cognition but controlled longitudinal studies are needed to confirm this, as practice effects could mask declines.

·       Clinical implications are discussed but more specific, actionable guidelines for providers could be offered.

·       Long-term durability of RNS effects need to be assessed through long-term follow up beyond initial postoperative period.

·       Limitations section acknowledges observational design but could be expanded on. Confounding factors, selection bias, and causal inferences need to be more directly addressed.

Conclusion

·       The conclusion that RNS and RATL produce similar results is an overgeneralization that lacks nuance, given the limitations of the study design and analyses. Caution is needed when making equivalency claims.

·       The statement that RNS patients had higher baseline seizure frequencies is not quantified. The magnitude of difference should be reported and statistically tested.

·       The authors attribute the lower rate of seizure freedom in RNS to higher baseline seizures, but other factors like electrode placement could also contribute. This causal claim requires more evidence.

·       Sweeping statements like "the RNS and RATL patients function well" require quantification. How was functioning measured? What metrics defined "well"?

·       Using results to "easily counsel patients" is an overreach given the limitations of an observational study with a small RNS sample. Conclusions about best options for patients require clinical outcome trials.

·       No concrete clinical recommendations are provided, despite the stated goal to help guide patient counseling. Specific, evidence-based advice is needed.

·       The statement about assisting patients in alignment with goals requires detailing how this study achieves that aim. Which results relate to which goals?

Author Response

(The authors gave the same response as above.)

Round 2

Reviewer 2 Report

Comments and Suggestions for Authors

The authors responded to my comments and concerns very well. Thank you.